# Bimanual Dexterity for Complex Tasks

**Kenneth Shaw**[*]     **Yulong Li**[*]

**Jiahui Yang**     **Mohan Kumar Srirama**     **Ray Liu**     **Haoyu Xiong**

**Russell Mendonca**[†]     **Deepak Pathak**[†]

Carnegie Mellon University

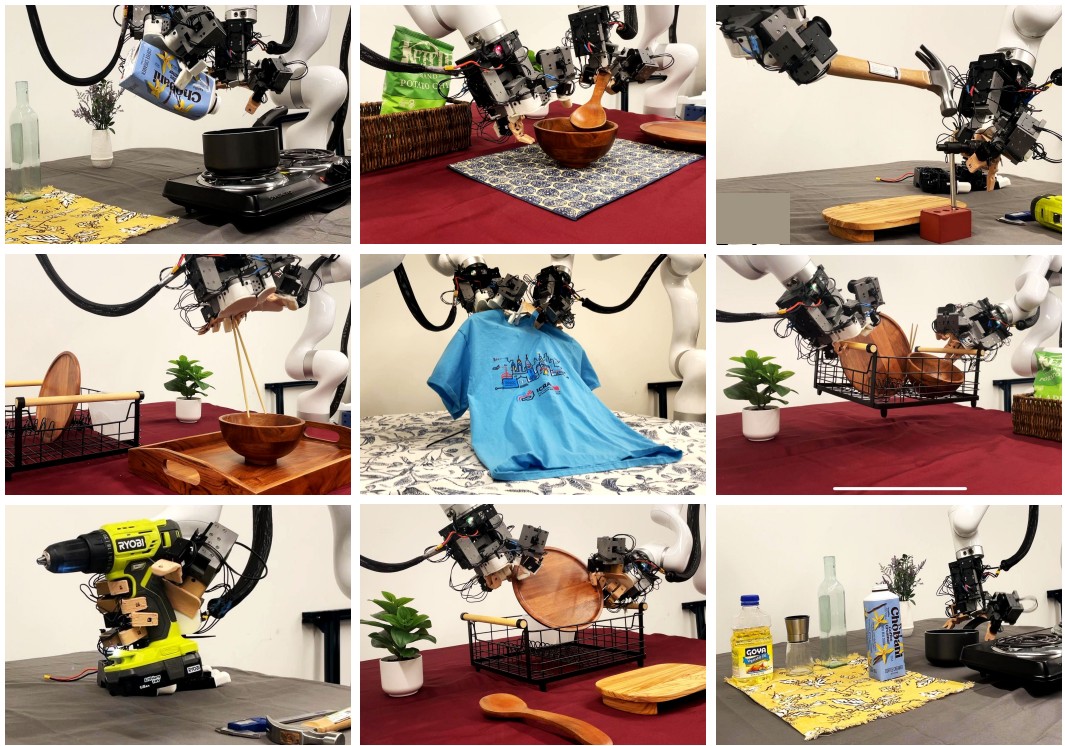

Figure 1: **Bimanual Dexterity**: BiDex can effortlessly teleoperate various complex tasks including pouring, scooping, hammering, chopstick picking, hanger picking, picking up basket, drilling, plate pickup and pot picking to train high-quality behavior cloning policies with over 50 degrees of freedom. Our teleoperation system and two LEAP hands [1] costs around $12k in total and is readily reproducible by academic labs.

**Abstract:** To train generalist robot policies, machine learning methods often require a substantial amount of expert human teleoperation data. An ideal robot for humans collecting data is one that closely mimics them: bimanual arms and dexterous hands. However, creating such a bimanual teleoperation system with over 50 DoF is a significant challenge. To address this, we introduce BiDex, an extremely dexterous, low-cost, low-latency and portable bimanual dexterous teleoperation system which relies on motion capture gloves and teacher arms. We compare BiDex to a Vision Pro teleoperation system and a SteamVR system and find BiDex to produce better quality data for more complex tasks at a faster rate. Additionally, we show BiDex operating a mobile bimanual robot for in the wild tasks. Please refer to https://bidex-teleop.github.io for video results and instructions to recreate BiDex. The robot hands ($5k) and teleoperation system ($7k) is readily reproducible and can be used on many robot arms including two xArms ($16k).

**Keywords:** Bimanual, dexterous hands, behavior cloning

---

[*]Equal contribution. [†] Equal advising.

8th Conference on Robot Learning (CoRL 2024), Munich, Germany.

# 1 Introduction

General-purpose robots in human environments will need to perform a wide variety of challenging manipulation tasks. This ranges from intricate movements like screwing in small objects, to cutting vegetables, to operating tools to being able to move large objects like furniture. These are tasks built around humans, and correspond to activities that people can perform. The versatility of human hands in particular is essential for finer-grained tasks ranging from writing and creating art to typing on keyboards. Hence one approach to building such versatile robot systems is to use a hardware form factor that resembles humans with two arms each equipped with a dexterous multi-fingered hands.

With the advent of data-driven machine learning methods and low-cost hardware there has been renewed interest in humanoids and dexterous hands. There is great promise for machine learning approaches to enable effective autonomous control for high-dimensional robot systems using large amounts of data [2, 3, 4, 5, 6]. A key question remains: how do we collect high-quality expert data for bimanual robots? Such a data collection system must be low-cost, easy to setup and use, low-latency and most importantly accurate enough. It should be effortless for the teleoperator to collect high quality data of robots performing complex tasks of interest to train robot policies.

To address this problem, VR headsets have become increasingly prevalent due to their easy-to-use internal body tracking systems [7, 8]. However we find that wrist tracking is often jittery and the finger tracking is inaccurate. To mitigate this, SteamVR [9] uses LiDAR which provides less noisy estimates, but requires external tracking devices which doesn't allow for data collection for mobile robot setups. For even higher fidelity readings, there is work that uses motion capture and reflective marker-based approaches such as Vicon or Optitrack[10] but they are extremely expensive and difficult to setup. An aspect of motion capture technology that has been used in robot learning in recent years are wearable gloves [11, 1, 12, 13] for hand tracking which records human fingertip position using EMF sensors. We use the accuracte Manus Meta glove as part of our system. [14]

For arm tracking, researchers in the robotics community have been recently using joint-level tele-operation for 2-finger grippers [15, 16]. Wu et al. [16]. They find that a low-cost 3D printed scaled teacher arm model that has the same kinematic link structure of a large robot arm can be used for accurate and effective teleoperation. These methods only provide one DOF of finger tracking instead of the twenty two plus DOF of the human hand. Our key insight is to develop a system that combines this joint-based arm tracking along with a Mocap fingertip glove to achieve accurate low-cost teleoperation of an arm and hand system.

Our contribution is BiDex, a system for dexterous low-cost teleoperation system for bimanual hands and arms in-the-wild in any environment. To operate, the user wears two motion capture gloves and moves naturally to complete everyday dexterous tasks. The gloves captures accurate finger tracking to map motions naturally to the robot hands. The GELLO [16] inspired arm tracking accurately tracks the human wrist position and joint angles for the robot arm. The data collected by the system can be used to train effective policies using imitation learning, after which the bimanual robot system can perform the task autonomously. BiDex costs around $6k for a pair of Manus gloves and few hundred dollars for the arm teleoperation which works for many existing robot arms. Even including the robot arms ($8k xArms x2) and robot hands used in our demonstration ($3k LEAP Hand V2 x2), the total cost is under $30k. We compare the accuracy and speed of data collection to other commonly used systems: the VR headset and SteamVR. We release videos of our results and instructions to recreate the setup on our website at https://bidex-teleop.github.io

# 2 Related Work

**Robot Arm Teleop** Many common approaches to teleoperation include using joysticks, space mouses [17], vision-based methods, [18, 19], and VR headsets [20, 8, 7] which control arms with inverse kinematics. Joint based teleoperated control has been used in areas such as kinesthetic teaching, Brantner and Khatib [21], ABB YuMi [22] and Da Vinci Machines [23], and recently [24, 25] introduced a low-cost version of this with mirroring Trossen Robot arms. GELLO [16] uses light and

inexpensive teacher arms that are 3D printed to control full size robot arms, a system we use in our BiDex due to its lost-cost, accurate, portable design.

**Robot Hand Teleop** The high dimensionality makes tracking human hands particularly difficult. To control robot hands, many vision-based techniques such as [26, 27, 19] do not require specialized equipment but are not that accurate. Shadow Hand developed a professional system that uses SteamVR and two gloves to control two Shadow Hands [28]. Recently, bimanual robot hands and tracking have become accessible for academic labs. Dexcap uses LEAP Hand [1] and tracks the human with gloves and a SLAM-based robot camera. [11] Hato uses a VR headset and controller to control two 6 DOF Psyonic Hands [29, 30]. A key question in controlling robot hands is how to map the human hand configuration to robot hand joints. These papers introduce inverse kinematics based methods that optimize pinch grasps between the human and robot hands [11, 26, 19, 27].

**Motion Capture** Motion capture and graphics contributions often are useful in the robotic teleoperation domain. Outside-in mocap approaches use external sensing technology to track the human body or other objects in the scene. SteamVR uses external lasers and worn wireless laser receivers. [9] Vicon-based systems use reflective balls and external cameras to track. Inside-out approaches such as XSens [31] or Rokoko suit rely on IMUs on the body but these often drift over time and require recalibration [32, 29]. For hand data, many vision-based approaches such as Frankmocap [33] return MANO [34] parameters which can be converted to robot hand joint angles.

**Learning from Expert Demonstrations** Recently the robot learning community has seen notable success in learning from demonstrations driven by the development of imitation learning algorithms [35, 36]. Complementing these advances, significant efforts have been made to scale up robotic datasets to facilitate more capable robotic systems [37, 38, 39, 40]. Despite these efforts, acquiring robotics data remains an expensive and challenging endeavor. To address these issues, developments in low-cost hardware have been instrumental in democratizing access to robotic technology, enabling more widespread research and application [41, 24, 25, 42]. However, these systems are primarily focused on simple gripper functionalities; and the challenge of achieving more intricate dexterity and intuitive control in robotic systems motivates our bimanual dexterous teleop system.

## 3 Bimanual Robot Hand and Arm System

We present BiDex, a system that allows any operator to effortlessly teleoperate a bimanual robot hand and arm setup. BiDex is designed to be exceptionally precise, affordable, low-latency, and portable, enabling control of any human-like pair of dexterous hands, even those with over 20 degrees of freedom. It achieves accurate tracking of the human hand using a Manus VR glove-based system [14] and human arm tracking through a GELLO-inspired system [16]. We outline the process for sending commands and collecting data with bimanual hands in Alg.1. Importantly, our solution functions seamlessly in both tabletop and mobile environments, as it requires no external tracking devices and is highly portable. In Section 5, we demonstrate that our system is highly intuitive, precise, and cost-effective compared to widely used methods today, such as VR headsets and SteamVR, across two different pairs of open-source robot hands, LEAP Hand [1] and LEAP Hand V2 [43].

### 3.1 Multi-fingered Hand Tracking

A hand tracking system must deliver accurate, low-latency joint information for the human hand, which has over 20 degrees of freedom. Many current vision-based tracking solutions, such as those using FrankMocap [33, 44] or VR headsets, often face significant inaccuracies due to occlusions and varying lighting conditions, as discussed in Section 5. In contrast, recent motion capture gloves that utilize EMF sensors provide significantly more accurate tracking without being overly expensive. They avoid the occlusion issues common in vision-based methods and offer detailed data on the skeletal joint structure of the human hand. Additionally, these gloves can be worn comfortably without hindering movement. In BiDex, we opt for the Manus Glove [14], which has demonstrated reliable tracking performance without overheating or suffering from calibration problems as seen with alternatives like the Rokoko gloves [32, 29]. However, mapping this data from the human hand

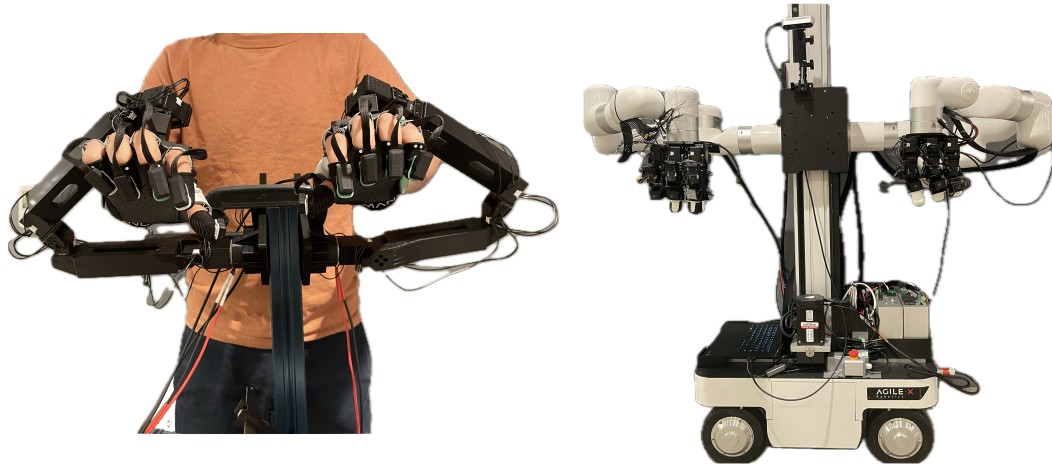

Figure 2: **Mobile bimanual teleoperation system** Left: An operator strapped into BiDex. Right: Our bimanual robot setup including two xArm robot arms, two LEAP Hands [1] and three cameras on an AgileX base.

to a robot hand remains challenging due to differences in their morphological structures. How can we translate commands from the operator's fingers to a robot hand that may have a different kinematic configuration?

If the kinematic structure of the robot hand is roughly human-like, one approach would be to directly map the joint angles from human finger joints to those of the robot hand. Although these gloves do not provide true joint angles, they can compute them using an inverse kinematics solver applied to a standard human skeleton. However, if the robot fingers differ significantly in size and proportions from human fingers, the resulting motions may not align properly. The human thumb, in particular, has a complex joint configuration that many robot hands fail to replicate accurately, complicating intuitive thumb control. This can lead to inaccuracies in pinch grasps, negatively impacting the reliability of task performance, as effective manipulation heavily depends on the relative positions of the fingertips.

Previous work [26, 19] has addressed this challenge by ensuring that pinch grasps are consistent between human and robot hands. Wang et al. [11] demonstrated that effective mapping can be achieved by optimizing the

---

**Algorithm 1** Teleoperation Data System

**Require:** Two robot arms $Al, Ar$
**Require:** Two robot multi-fingered hands $Hl, Hr$
**Require:** Kinematic models for two arms $Kl, Kr$
**Require:** Gloves with fingertip trackers $Gl, Gr$
**Require:** Robot hand IK model for fingertips $Q_{l,r}()$
**Require:** RGB workspace cameras $\{I_c\}$
**Require:** Number of trajectories to be collected $N$
1: Initialize Data buffer $\mathcal{D}$
2: **for** trajectory 1:N **do**
3:    Initialize trajectory $\mathcal{T} = \{\}$
4:    **while** task not completed **do**
5:      Read camera images $\{I_c\}_t$
6:      Read arm joints $Al_t, Ar_t$
7:      Read robot hand joints $Hl_t, Hr_t$
8:      Observation $o_t = \{Al_t, Ar_t, Hl_t, Hr_t, \{I_c\}_t\}$

9:      Read kinematic arm model joints $Kl_t, Kr_t$
10:     Read glove fingertip positions $Gl_t, Gr_t$
11:     Finger joints $ql_t = Q_l(Gl_t), qr_t = Q_r(Gr_t)$
12:     Action $a_t = \{Kl_t, Kr_t, ql_t, qr_t\}$
13:     Add $(o_t, a_t) \mapsto \mathcal{T}$

14:     Set joints of $Al$ using $Kl_t$ and $Ar$ using $Kr_t$
15:     Set joints of $Hl$ using $ql_t$ and $Hr$ using $qr_t$
16:   **end while**
17:   Add $\mathcal{T} \mapsto \mathcal{D}$
18: **end for**
19: **return** Data buffer $\mathcal{D}$

---

joint positions of the fingertip and penultimate joint (DIP) on each finger in relation to the wrist, ensuring similarity between the human and robot hands through an SDLS IK solver [45]. We employ the Manus gloves [14] using a similar inverse kinematics approach, which allows for precise pinch grasps and proper thumb positioning.

## 3.2 Arm Tracking

Our system must accurately track the human wrist pose to control two robot arms. Traditionally, various methods have been developed by both the motion capture and robotics communities. However, many of these approaches rely on calibrated external tracking devices which either are costly or have high latency. These external tracking devices are also non-portable, making it hard to scale to mobile systems. Instead, we leverage key insights from Zhao et al. [15], Wu et al. [16] which both use lighter teacher arms attached to the human arm to control a robot arm and hand system. Specifically we follow the GELLO system from Wu et al. [16] to teleoperate a full size robot arms. A key question is how to mount this arm-tracking system on a human wrist and hand. If the robot hand is mounted on the arm in a human-like way, then the glove needs to be mounted in a human-like way on the teacher arm to match. However, this orientation means that the human arm will be parallel with the teacher arm and constantly collide with it. This is jarring for the operator and uncomfortable.

In BiDex, we mount the robot hands underneath the arms in the same orientation as if it were a gripper as in Figure 2. When mirroring this in the teacher arm, the human arm and teacher arm output are perpendicular to each other and do not collide which is more comfortable. Because of the weight of the motion capture glove, we must adjust the teacher arm to be more robust. This includes adding a strong bearing to the base joint of the teacher arm and adding rubber bands to bias additional joints back to the center of the joint range.

## 3.3 Robot Configurations

**Tabletop Manipulation** For our tabletop setup, the robotic arms are positioned to face each other similar to Zhao et al. [15] while the GELLO teaching arms mirror this configuration. Compared to a side-by-side configuration like in Wang et al. [11], this setup has three main benefits: 1) the human operators avoid collisions with the teacher arms; 2) the setup allows better visibility of the workspace, which will be otherwise occluded by a side-by-side robot arm configuration; 3) and finally, the robot arms have a wider shared workspace.

**Mobile Manipulation** BiDex operates without the need for external tracking systems and is lightweight, making it well-suited for mobile settings. The teacher arms are mounted on a compact mobile cart. For the mobile robot, we attach two robot arms to an articulated torso capable of moving upward to reach high objects and downward toward the ground, similar to the PR2 [46], as shown in Figure 2. The robotic assembly, which includes the arms and torso, is mounted on an AgileX Ranger Mini, allowing for movement in any SE(2) direction [47, 48]. A secondary operator uses a joystick to control the mobile base and manage task resets.

# 4 Experiment Setup

## 4.1 Baseline Teleoperation Approaches

**Vision based VR Headset** In recent years, the accessibility of low-cost VR headsets using multi-camera hand tracking has made them popular for teleoperation such as in [7, 8] As a baseline we use the Apple Vision Pro which returns both finger data similar to MANO parameters [44] and wrist coordinate frame data. The finger data is used in the same way as with BiDex through inverse kinematics-based retargeting and commanded onto the robot hands. The wrist data is reoriented, passed through inverse kinematics and the final joint configuration is commanded to the arms.

**SteamVR Tracking** SteamVR, commonly used in the video gaming community has also seen recent interest in the robotics community from industry [28] and academia alike. [12, 49] It uses active powered laser lighthouses that must be carefully placed around the perimeter of the workspace. Wearable pucks with IMUs and laser receivers are worn on the body of the operator. In our experiment the operator wears one tracker on each wrist and one tracker on their belly. The wrist position is determined relative to the belly pose, mapped to the robot arm, and the joint angles are computed using inverse kinematics. The hand tracking gloves are the same as in BiDex.

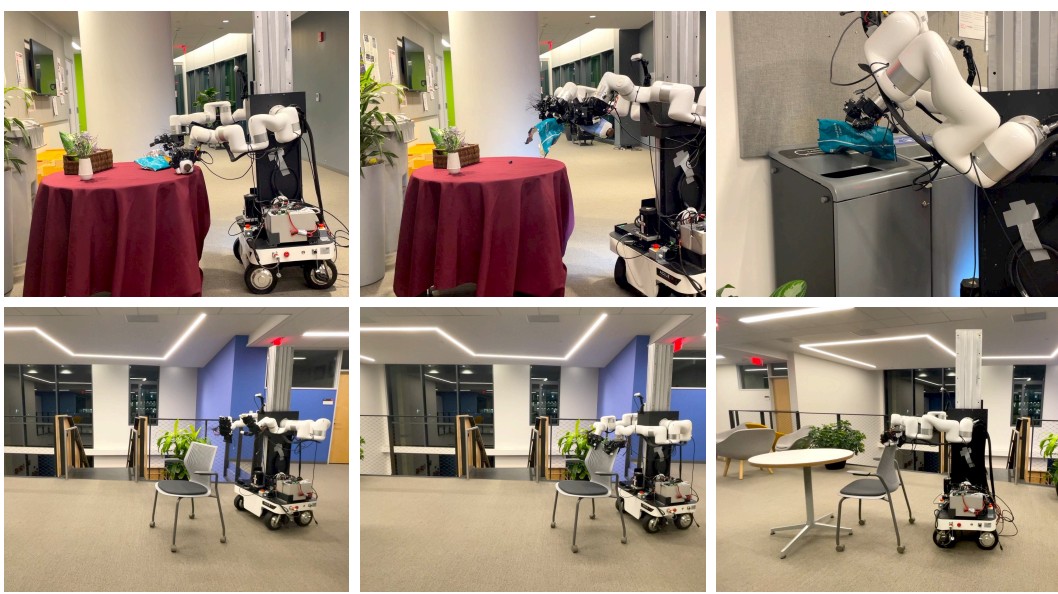

Figure 3: **All Tasks**: Teleoperation of the mobile robot systems with BiDex. **Top**: Picking up trash from a table and discarding it into a bin. **Bottom**: Grasping a chair and moving it to align with a table.

## 4.2 Choice of Dexterous End-Effectors

**Leap Hand** LEAP Hand, introduced by Shaw et al. [1] is a low-cost, easy-to-assemble robot hand with 16 DOF and 4 fingers. LEAP Hand introduces a novel joint configuration that optimizes for dexterity as well as human-like grasping. We use this hand for many experiments in the paper as it is a readily available dexterous hand available for comparison studies.

**LEAP Hand V2** We would like a hand that is smaller and more compliant than LEAP Hand. LEAP Hand V2 is crafted to mimic the suppleness and strength of the human hand with fingers that have a 3D-printed flexible outer skin paired with a sturdy inner framework resembling bones. These fingers do not break but instead bend and flex upon impact. We also introduce an active articulated palm which integrates two motorized joints, one spanning the fingers and another for the thumb, enabling natural tight grasping. LEAP Hand V2 contains 21 degrees of freedom and is sized to resemble a human hand, easy to assemble and is economical. Because of the human-like size and kinematics, it is easy to retarget to and can complete many more dexterous tasks successfully.

## 4.3 Task Descriptions

**Tabletop** In *Handover*, the robot picks up a pringles can and passes it from its right hand to its left hand in the air. In *Pour*, the robot pours from a glass bottle in one hand into a plastic cup held by the other hand. In *Tabletop Cup Stack* the agent stacks one cup into another cup in the other hand.

**Mobile** In *Transport Box*, the agent moves a box from one table to another table using two hands. In *Mobile Chair Push*, the agent needs to grasp a chair, and then align it with a table. In *Clear Trash*, the robot clears trash from the table into a dustbin. We visualize the chair push and clear trash tasks in Figure 3.

## 5 Results

We investigate BiDex teleoperating in both the tabletop scenario and in the mobile in-the-wild scenario against a few baselines. For these comparisons, we use LEAP Hand by Shaw et al. [1] because it is an open source easy to assemble robot hand that is readily attainable by any robotics lab. We also use BiDex with the more recent LEAP Hand v2, which is made of a combination of rigid and soft material and capable of performing more complex tasks.

|  | Completion Rate | | | Time Taken | | |
|---|---|---|---|---|---|---|
|  | Handover | Cup Stacking | Bottle Pouring | Handover | Cup Stacking | Bottle Pouring |
| Vision Pro VR | 60 | 40 | 70 | 21.6 | 38.8 | 35.5 |
| SteamVR | 80 | 85 | 60 | 17.5 | 16.5 | 15.5 |
| **BiDex** | 95 | 75 | 85 | 6.5 | 15.5 | 14.9 |

Table 1: **Tabletop Teleoperation**: We compare BiDex on the handover, cup stacking, and bottle pouring tasks to two baseline methods, SteamVR and Vision Pro. BiDex enables more reliable and faster data collection, especially for harder tasks like bottle pouring.

|  | Completion Rate | | | Time Taken | | |
|---|---|---|---|---|---|---|
|  | Chair Pushing | Box Carry | Clear Trash | Chair Pushing | Box Carry | Clear Trash |
| Vision Pro VR | 75 | 75 | 50 | 15.0 | 33.7 | 79.8 |
| **BiDex** | 95 | 95 | 75 | 16.4 | 29.7 | 74.6 |

Table 2: **Mobile Teleoperation**: Completion rate and time taken averaged across 20 trials using a mobile bimanual system with LEAP Hand [1], for different tasks. BiDex is versatile and compact enough to be adopted to successfully collect data for mobile tasks.

## 5.1 Bimanual Dexterous Teleoperation Results

**BiDex provides more stable arm tracking.** The teacher arm system makes BiDex highly reliable, with minimal jitter, low latency and high uptime, making it ideal for arm tracking. The teacher arm is lightweight and doesn't impede the user more than the gloves' weight. The kinematic feedback from arm resistance is subtle but helps operators navigate around arm singularities intuitively. As shown in Tables 1 and 2, BiDex achieves a higher completion rate in less time for teleoperators.

In contrast, the Vision Pro often experiences jittery arm tracking, complicating the teleoperation of more demanding tasks. While low-pass filtering can mitigate this issue somewhat, it introduces undesirable latency. Occasionally, the system may stop functioning entirely, which can be disconcerting for users.

The SteamVR system offers wireless connectivity, allowing users to be untethered, and it generally provides accurate tracking. However, it can experience brief episodes of high latency or disconnections every 5-10 minutes, which can be jarring. Notably, the SteamVR system cannot be used in mobile settings due to the need for external tracking lighthouses to be set up around the teleoperation environment.

**BiDex provides more accurate hand tracking.** With BiDex, fingertip tracking is highly accurate when using the Manus glove. When mapping to different robots, only minor adjustments to inverse kinematics are required for operators with varying hand sizes. The accuracy of abduction and adduction at the MCP joint remains dependable across different conditions. These benefits are particularly crucial in LEAP Hand V2, where precision is essential for executing more complex tasks.

In contrast, the Vision Pro can struggle with hand size variability under different lighting conditions, making the retargeting process to robot hands more challenging. Additionally, finger abduction and adduction estimates can be impacted by occlusions, which complicates the performance of intricate tasks. The latency is noticeable and can pose a significant issue for teleoperation.

|  | Can Handover | Cup Stacking | Bottle Pouring |
|---|---|---|---|
| Leap Hand | 7/10 | 14/20 | 16/20 |

Table 3: **Imitation learning**: We train ACT from [15] using data collected by BiDex and find that our system can perform well even in this 44 dimension action space. This demonstrates that our robot data is high quality for training robot policies.

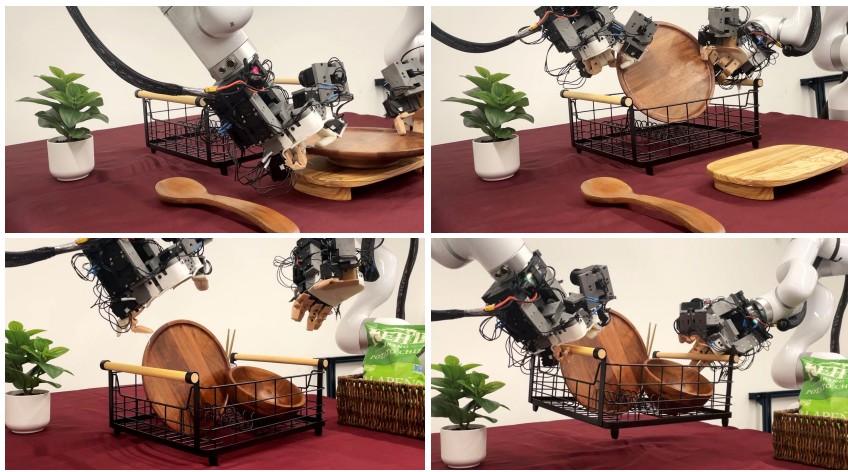

Figure 4: **Clearing the Dishes**: In this task, we use BiDex to perform a long horizon task to place bowls and spoons into a drying rack and lift the drying rack away from the table.

## 5.2 Training Dexterous Visuomotor Policies with BiDex

To verify that the data that is collected by our system is high quality and useful for machine learning we train single task closed loop behavior cloning policies.

Specifically, we train an action chunking transformer from [15] with a horizon length of 16 at 30hz using pretrained weights from [50] on around 50 demonstrations. The state space is the current joint angles of the robot hand and the images from the camera. The action space in the case of LEAP Hand is 16 dimensions for each hand and 6 dimensions for each arm for a total of 44 dimensions. During rollouts, the behavior of the policies are very smooth, exhibiting the high quality of the teleop data. In tasks such as the YCB Pringles can [51] handover, we even see good generalization of the policy to different initial locations of the can.

## 5.3 Extreme Dexterity using LEAP Hand V2

To push BiDex to its dexterous limits, we use LEAP Hand V2: an extremely dexterous 21 DOF hybrid low-cost hand which is explained in Section 4.2. To do this, we show a variety of very challenging tasks as in Figure 1 and Figure 4. These tasks include pouring, scooping, hammering, chopstick picking, hanger picking, picking up basket, drilling, plate pickup and pot

|  | Drill | Lift Pot | Bottle Pouring | Plate Pickup |
|---|---|---|---|---|
| Leap Hand v2 | 15/20 | 15/20 | 15/20 | 13/20 |

Table 4: **Imitation learning LEAP v2**: We also train ACT using LEAP Hand v2 and show task completion on more dexterous tasks.

picking. In these experiments we find that BiDex scales well to this high DOF hand and it feels very natural to control this soft-rigid robot hand. We provide Table 4 and video results on our website at https://bidex-teleop.github.io

## 6 Discussion and Limitations

In this paper, we introduce BiDex, a portable, low-cost and extremely accurate method for teleoperating a bimanual, human-like robot hand and arm system. We demonstrate the system's applicability to both a tabletop and a mobile setting and show its efficiency in performing bimanual dexterous tasks in comparison to alternative approaches including SteamVR and Vision Pro. Nevertheless, our BiDex is not without limitations. Due to the lack of haptic feedback, the human operator has to rely on visual feedback for teleoperation and cannot feel what the robot hand is feeling. Additionally, they cannot exert intricate force control and can only control the kinematics of the robot hand and arm which can make it challenging for fine-grained manipulation tasks. A promising direction in the future would be to integrate haptic feedback into our system which will unlock further potential for collecting extreme dexterity data.

**Acknowledgments**

We thank Ankur Handa, Arthur Allshire, Toru Lin and Nancy Pollard for discussions about the paper. We also thank Maarten Witteveen and Sarah Shaban at Manus Meta for their assistance on their gloves. We thank Diya Dinesh and Lukas Kebuladze for helping with teleoperation. This work is supported in part by DARPA Machine Commonsense grant, AFOSR FA9550-23-1-0747, ONR N00014-22-1-2096, AIST CMU grant and Google Research Award.

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

# Supplementary: Bimanual Dexterity for Complex Tasks

**Anonymous Author(s)**
Affiliation
Address
`email`

## 1  Videos, Assembly Instructions and Software on our Website

Please see our project website for videos, assembly instructions and software. This information is useful to recreate BiDex and create variants of it using high quality motion capture gloves.

## 2  Detailed Cost Analysis

Please see Table 1 and Table 2 for a detailed Bill of Materials and breakdown of the cost to create BiDex. This is accurate pricing as of the paper submission. While we assert that BiDex is low cost, we acknowledge that it is still not affordable for everyone such as hobbyists. We believe that the price of motion capture gloves will continue to decrease over time as technology improves and demand increases in our field as well as other adjacent fields. Altogether, the cost of the robot arms, hands, and teleop system costs around $30k and can be accessible for academic or industry labs.

| Object | Quantity | Total |
|---|---|---|
| Pair of Manus Meta Gloves | 1 | $6000 |
| Dynamixel XL330-M288 (Gello) | 12 | $300 |
| U2D2 Control PCB | 2 | $40 |
| 5v 20A Power Supply | 2 | $25 |
| 14 AWG Cabling | 1 | $20 |
| PLA Printer Plastic | N/A | $10 |
| Total | | $6395 |

Table 1: We present the bill of materials of BiDex for two tracker arms and gloves. The total cost is around $6000, mostly due to the Manus Meta gloves.

| Object | Quantity | Total |
|---|---|---|
| xArm 6 | 2 | $18000 |
| Ubuntu Laptop | 1 | $2000 |
| Mobile Base | 1 | $6000 |
| Zed Camera | 3 | $1200 |
| LEAP Hand or LEAP Hand V2 | 2 | $4000 |
| Total | | $31,2000 |

Table 2: We present the bill of materials of the mobile robot setup. The robot and BiDex costs around $35,000 in total which we believe is reasonable for a dexterous bimanual robot hand setup with 50+ degrees of freedom.

# 3 User Study

To further evaluate BiDex, we conducted a user study involving novice users who tested both BiDex and the Apple Vision Pro system. Each participant performed the Pringles handover task over 10 trials, following a 3-minute practice session with each system. We collected data on average task completion times and success rates, as well as users' ratings (on a scale from 1 to 5) regarding accuracy, responsiveness, ease of use, and their confidence in each system. The results are summarized in Fig. 4.

Our findings reveal that users completed tasks significantly faster with BiDex. While all participants achieved high success rates with BiDex, the Apple Vision Pro exhibited greater variability in performance. Notably, one user (User 5) struggled with controlling the Apple Vision Pro, resulting in a broken robot hand and a zero success rate for that trial. However, a few users managed to achieve success rates with the Apple Vision Pro that were comparable to those with BiDex. Overall, participants rated BiDex as more accurate, responsive, and easier to use, and they expressed greater confidence in using it.

# 4 Policy Performance Comparison

We present additional results to compare and evaluate policies trained from data collected with BiDex and the Apple Vision Pro. In particular, we collected 100 demonstrations on the Pringles can handover task using both systems, and trained policies with different numbers of demonstrations. We then evaluate each policy for 10 trials at roughly the same starting poses. We summarized our results in Fig. 2.

In general, we found that policy performance scales with the number of demonstrations, an unsurprising result which highlights the need of efficient and effective teleoperation systems to collect large amounts of robotic data. Before conducting the evaluation, we hypothesized that policies trained on different data sources would achieve similar performance given the same number of demonstrations. However, to our surprise, policies trained from the Apple Vision Pro perform worse, mainly due to abrupt wrist movements, as shown in the "Apple Vision Pro vs Ours" section. We hypothesize that the difference in action spaces between two teleoperation systems results in the performance differences. The Apple Vision Pro commands in end-effector space, and a small prediction error can result in large errors in joint space. While BiDex commands directly in joint space, which results in smoother actions.

# 5 About Manus Glove

We use the Manus Meta Quantum Metagloves [1] which is an $6000 tracking Mocap glove. Each finger is tracked by the glove and returns the fingertip positions as xyz-quaterion and also 4 different angles for each finger $\theta_{MCP_{side}}, \theta_{MCP_{fwd}}, \theta_{PIP}, \theta_{DIP}$ using hall effect sensors with very high accuracy and at 120hz. We use their Windows API (Linux is not available at time of release) and release our

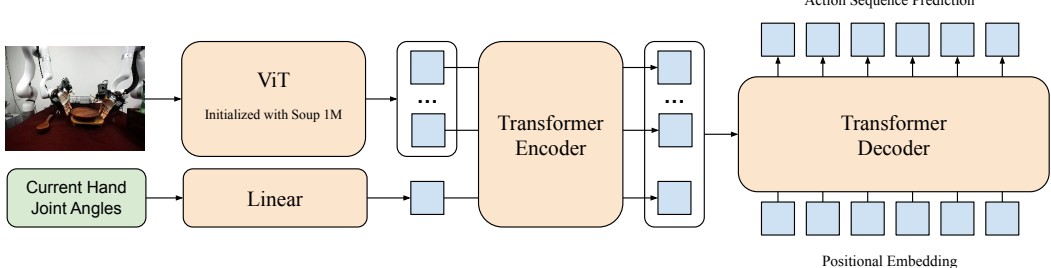

Figure 1: Behavior Cloning Policy Architecture

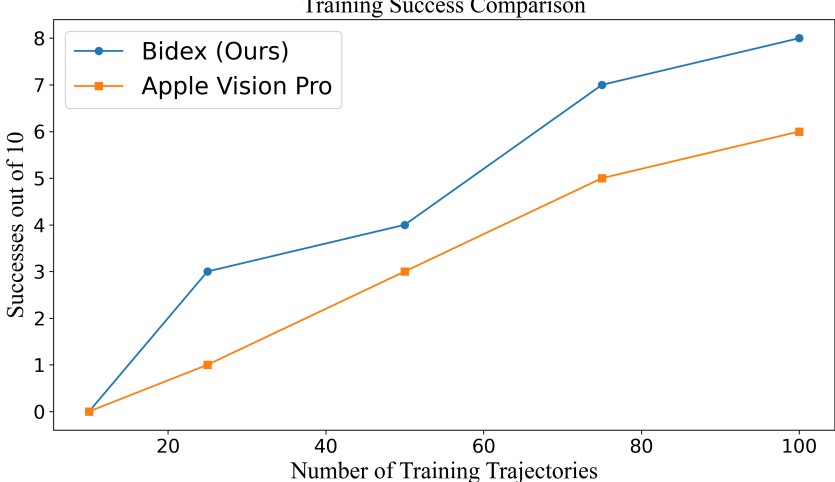

Figure 2: Policy Performance with varying number of demonstrations collected with BiDex and the Apple Vision Pro.

version of that which sends the software to a Linux machine running ROS. These gloves are available for purchase at https://www.manus-meta.com/ and our software is available on our project website

# 6    SteamVR Baseline

For the wrist tracking SteamVR baseline, we use the Manus Meta SteamVR trackers which connect to the gloves and seamlessly route the data through the aforementioned Windows API. They are wireless but require SteamVR Lighthouses setup around the perimeter of the workspace. In our test we mount the 4 SteamVR trackers on the ceiling to avoid as many occlusions as possible. We also mount the 4 trackers in a 16ft square around where the teleoperator would stand which is the recommended configuration. We will release this code for others to recreate in their comparison study.

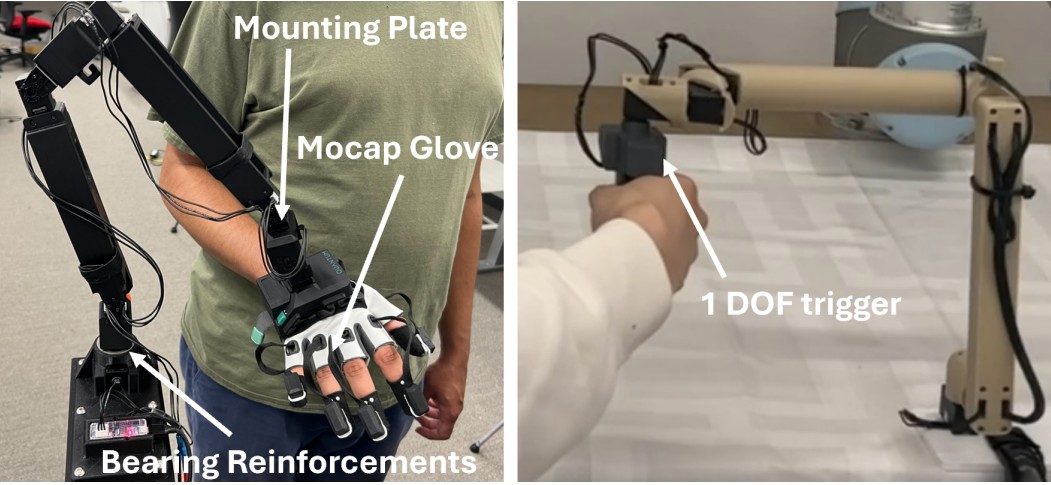

Figure 3: We compare BiDex to the GELLO system designed for 1 DOF grippers. Our system uses a motion capture glove to capture full fingertip information and is reinforced to handle the wears and tears of this additional weight.

| Hyperparameter | Value |
|---|---|
| **Behavior Policy Training** | |
| optimizer | AdamW |
| base learning rate | 3e-4 |
| weight decay | 0.05 |
| optimizer momentum | $\beta_1, \beta_2 = 0.9, 0.95$ |
| batch size | 64 |
| learning rate schedule | cosine decay |
| total steps | 10000 |
| warmup steps | 500 |
| augmentation | GaussianBlur, Normalize, RandomResizedCrop |
| GPU | RTX4090 (24 gb) |
| Wall-clock time | $\sim 1$ hour |
| **Visual Backbone ViT Architecture** | |
| patch size | 16 |
| #layers | 12 |
| #MHSA heads | 12 |
| hidden dim | 768 |
| class token | yes |
| positional encoding | sin cos |
| **Action Chunking Transformer Architecture** | |
| # encoder layers | 6 |
| # decoder layers | 6 |
| #MHSA heads | 8 |
| hidden dim | 512 |
| feedforward dim | 2048 |
| dropout | 0.1 |
| positional encoding | sin cos |
| action chunk | 100 |

Table 3: Hyperparameters for Behavior Cloning Policy Training

## 7 Apple Vision Pro Baseline

The Apple Vision Pro baseline is based off of [2]. With this data, we control the hand using the same inverse kinematics as with the Manus Glove. For the arm, we scale, translate and rotate for the robot embodiment and then pass through inverse kinematics to control the arm.

## 8 Behavior Cloning Policy Architecture and Hyperparameters

We illustrate our policy architecture in Figure 1. Our behavior cloning policy takes as input a RGB image and current hand joint angles (proprioception). We obtain tokens for the image observation via a ViT [3] and a token for joint proprioception via a linear layer. The weights of ViT is initialized from the Soup 1M model from [4]. The tokens then pass through a action chunking transformer [5], a encoder-decoder transformer, to output a sequence of actions. The action space is the absolute joint angles of two arms and two hands. A key decision that greatly improves policy generalization is to exclude current arm joints from the proprioception. Intuitively, this may force the model to extract object information from image observations, rather than overfitting to predict actions close to current arm states.

We list key hyperparameters for our behavior policy training Table 3. In general, we are able to obtain well-performing policies with 20-50 demonstrations and 1 hour of wall-clock time training on a RTX4090. With our easy-to-use teleoperation system, we are able to obtain diverse policies for complex bimanual dexterous tasks quickly.

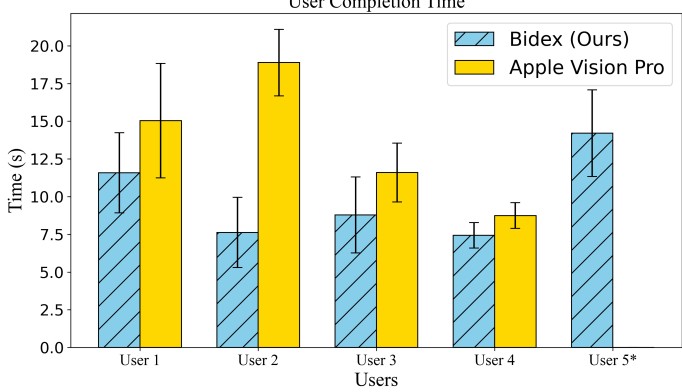

(a) User mean and standard deviation completion times.

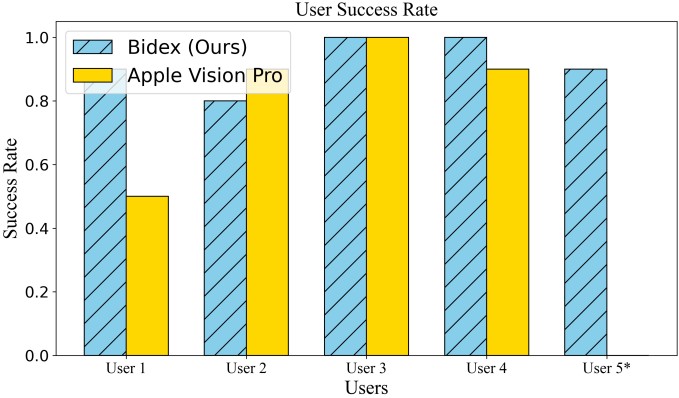

(b) User success rates.

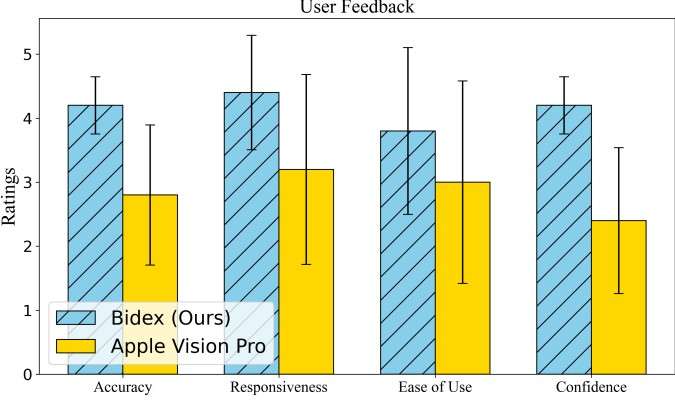

(c) User feedback on accuracy, responsiveness, ease of use, and confidence in using each system.

Figure 4: Novice operators were asked to complete the Pringles can handover tasks for ten trials with BiDex and the Apple Vision Pro. * User 5 found it hard to control the system with the Apple Vision Pro and broke robot hands during operation, resulting in zero success rate.

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
