# OpenReview forum: "Bimanual Dexterity for Complex Tasks"
_robot-learning.org/CoRL/2024/Conference — CoRL 2024_

### Official Review · Reviewer_kK4n · 2024-07-12
**Good paper with very little learning**

**Originality:** 3
**Technical Quality:** 4
**Clarity Of Presentation:** 4
**Potential Impact:** 3
**Recommendation:** 2
**Confidence:** 3

**Review:**

The paper is well written and the presented hardware seems to be a valuable contribution to the robot learning community. The design decisions made by the authors are well explained and sound reasonable to me. The device is affordable (~5k$) and seems to allow for accurate wrist pose and finger pose tracking. Related work seems to be cited in in an adequate way. The paper is a hardware paper that uses learning only as a proof of concept. That means the authors showcase that their hardware can be utilized for collecting data that can be used for training imitation learning models. It really is a good paper but I don't think CoRL is the right conference for it because the authors do not solve any learning problem. A conference like ICRA (or another conference that is more general or has a focus on robotics hardware) is a much better fit for the paper but it's up to the Area Chair to decide this.

More minor weaknesses/suggestions:
- I felt like Section 3.2 was a bit too short. More details on the differences to GELLO (e.g., a picture comparison) would improve the paper.
- The authors could have discussed if it’s possible to use other end-effectors on the robot and teacher side. If my robot has only grippers, the Manus gloves are not really necessary and could be replaced by a cheaper input device. What would be necessary to modify the setup for controlling robots that have only grippers?

**Quality Of The Limitations Section:**

3

**Questions For Rebuttal:**

- The video on the homepage does not play. I tested it in three different browsers (firefox, chrome, edge). Only downloading the video and then playing it in VLC seems to work. Please re-check the encoding of the video.

- Typo: "biamanual" in caption of Figure 2

**Robotics Focus:**

4

**Summary Of Paper:**

The authors present a low-cost bi-manual teleoperation device which is comprised of two teacher arms for tracking the operator's wrist pose and Manus gloves for tracking the operator's finger poses. The device can be mounted to a table or on a mobile base, e.g., for teleoperating a mobile robot. In the latter case a second operator must control the pose of the mobile base. The device can be utilized for teleoprating a dexterous bimanual robot and/or to collect demonstration data, e.g., for imitation learning approaches.

**Summary Of Recommendation:**

As I said, it is a good paper but I don't think CoRL is the right conference for it because the focus of the paper is not on learning but on actual hardware that might be of interest to the learning community.

---

### Official Review · Reviewer_vkX4 · 2024-07-20
**Bimanual Dexterity for Complex Tasks**

**Originality:** 1
**Technical Quality:** 3
**Clarity Of Presentation:** 4
**Potential Impact:** 2
**Recommendation:** 3
**Confidence:** 3

**Review:**

Strengths:
1. The proposed BiDex system addresses an important need for affordable, accurate teleoperation of high-DOF bimanual robot systems.
2. The evaluation includes both tabletop and mobile manipulation scenarios, showing the versatility of the system.
3. The authors demonstrate the utility of the collected data by training imitation learning policies that achieve reasonable performance.

Weaknesses:
1. The technical novelty is somewhat limited - the key components (motion capture gloves, teacher arms) have been used in prior work. While the integration is valuable, the paper could benefit from a clearer articulation of the novel technical challenges overcome in combining these components for bimanual control. More discussion of the design choices and engineering challenges would strengthen the contribution.
2. The evaluation is primarily focused on task completion rates and times. Quantitative measures of tracking accuracy, latency, and precision of object manipulation would provide a more complete picture of the system's capabilities. Additionally, a user study with participants not involved in the system's development would help validate the claimed intuitiveness and ease of use.
3. The imitation learning results are preliminary - more extensive evaluation and comparison to other policy/data acquisition methods would be valuable. While the reported success rates are promising, more comprehensive experiments comparing to other learning approaches or data collection methods would better contextualize the contribution. Learning curves, analysis of policy generalization, and comparisons to policies trained on data from other teleoperation systems would significantly strengthen this aspect of the paper.

**Quality Of The Limitations Section:**

3

**Questions For Rebuttal:**

1. Can you provide more detailed analysis of the precision/accuracy of the arm and hand tracking, beyond just task completion metrics?

2. What are the primary failure modes or limitations of the BiDex system? Are there certain types of tasks or motions that are particularly challenging?

3. The paper mentions that BiDex enables "extreme dexterity", but the tasks shown are fairly standard manipulation. What is precenting more complex dexterous tasks that truly push the limits of the system?

**Robotics Focus:**

4

**Summary Of Paper:**

This paper introduces a low-cost portable teleoperation system for bimanual dexterous manipulation using robot arms and hands. The key components are motion capture gloves for hand tracking and lightweight "teacher arms" for arm tracking. The authors compare BiDex to baseline teleoperation methods using VR headsets and SteamVR, demonstrating improved performance on several manipulation tasks in both tabletop and mobile robot settings. They also show the system can be used to collect data for training imitation learning policies.

**Summary Of Recommendation:**

Overall, this paper presents a useful system integration that could benefit the robot learning community by enabling more widespread collection of dexterous manipulation data. While the technical novelty is not extremely high, the potential impact and practical utility of the system make this a valuable contribution. The paper would be strengthened by more rigorous evaluation and analysis.

---

### Official Review · Reviewer_ew2f · 2024-07-21
**Cool new hardware tools for the dexterous robotics community**

**Originality:** 2
**Technical Quality:** 2
**Clarity Of Presentation:** 4
**Potential Impact:** 4
**Recommendation:** 4
**Confidence:** 2

**Review:**

The paper is well-written and clearly explained. The figures and video demonstrations are quite convincing in demonstrating the need for two-arm multi-finger teleoperation systems. The authors showcase the system's versatility through both tabletop tasks (handover, cup stacking, bottle pouring) and mobile manipulation scenarios (chair pushing, box carrying, trash clearing).

The authors claim that BiDex is "exceptionally precise", but don't offer quantitative data to support this claim. They also assert that using the system is "highly intuitive"; some user feedback studies could be useful in supporting this claim.

The experimental results, while promising, are quite limited. The paper would be significantly improved with additional precision and latency testing, as well as statistical analysis of the results. The results table doesn't report standard deviations or perform a statistical analysis of the outcomes. Crucially, the paper doesn't specify who is operating the system during these experiments. It's very important that it's not the authors but a selected group of participants to avoid potential bias.

In terms of scientific novelty, the paper is somewhat limited, as it mainly assembles pieces from existing works (motion capture gloves, teacher arms).

**Quality Of The Limitations Section:**

3

**Questions For Rebuttal:**

See above

**Robotics Focus:**

4

**Summary Of Paper:**

The authors present BiDex, a low-latency, low-cost, and portable bimanual dexterous teleoperation system. The system combines motion capture gloves for hand tracking with lightweight teacher arms for arm control, offering an alternative to existing methods like SteamVR and Vision Pro.

**Summary Of Recommendation:**

Overall, in combination with the promised release of CAD models and source code, I think this paper would be valuable to the robotics community, despite its limitations. It provides a practical solution to bimanual dexterous teleoperation that could accelerate research in this area. However, the authors should address the lack of rigorous validation for their claims and expand their experimental analysis to strengthen the paper's contribution.

---

### Author Rebuttal · Authors · 2024-08-08

Please see the individual comments to the Meta-Reviewer and the individual Reviewers for the rebuttal.  The file attached to this rebuttal message is for additional experiments and figures that were requested by the three reviewers.

---

### Decision · Program_Chairs · 2024-09-04

**Decision:**

Accept

**Comment:**

**Pre-rebuttal**

This paper receives a mixed review from three reviewers. Overall, all three reviewers acknowledge the potential value of the hardware system to the robot learning community.

Strength mentioned:
- Strong relevance (**vkX4**).
- Versatility of the system (**vkX4**, **ew2f**).
- Demonstration of the utility of the collected data (**vkX4**).
- Clarity of presentation (**kK4n**, **ew2f**).

Weaknesses mentioned:
- Limited / unclear technical novelty (**vkX4**, **ew2f**).
- Lacking in evaluation quality (e.g. accuracy, latency, etc.) (**vkX4**, **ew2f**).
- Evaluation with lay users (**vkX4**, **ew2f**).
- Lacking in comparison with other imitation learning approaches / data collection methods (**vkX4**).
- Topic fit to CoRL (**kK4n**).

---
**Post-rebuttal**

This paper initially received one strong accept, one weak accept, and one weak reject. All the reviewers retain their original rating after the rebuttal.

After reading the paper, reviews, and the rebuttal, AC considers this paper a borderline case. Particularly, AC acknowledges the merits of the hardware contribution and the substantial engineering effort for putting together the system. Meanwhile, AC agrees with following shortcomings:
- Lack of rigorous evaluation.
- Lack of scientific novelty.
- Limited imitation learning results.

However, considering the recent research landscape and the growing interest in the teleoperation of bimanual dexterous setups, AC recommends to accept the paper with the belief that the paper will be of sufficient interest to the CoRL audience.

AC would also like to remark the importance of the user study and encourage the authors to include the new user study (from the rebuttal) to the main paper if the paper is eventually accepted.